# Preparation of Spherical Ultrafine Silver Particles Using Y-Type Microjet Reactor

**DOI:** 10.3390/ma16062217

**Published:** 2023-03-10

**Authors:** Xiaoxi Wan, Jun Li, Na Li, Jingxi Zhang, Yongwan Gu, Guo Chen, Shaohua Ju

**Affiliations:** 1Faculty of Metallurgical and Energy Engineering, Kunming University of Science and Technology, Kunming 650093, China; 2Key Laboratory of Unconventional Metallurgy, Ministry of Education, Kunming 650093, China; 3National Local Joint Laboratory of Engineering Application of Microwave Energy and Equipment Technology, Kunming 650093, China; 4Kunming Institute of Precious Metals, Kunming 650106, China

**Keywords:** Y-type microjet reactor, spherical ultrafine silver particles, wet chemical reduction, dendritic particle

## Abstract

Herein, micron-sized silver particles were prepared using the chemical reduction method by employing a Y-type microjet reactor, silver nitrate as the precursor, ascorbic acid as the reducing agent, and gelatin as the dispersion at room temperature (23 °C ± 2°C). Using a microjet reactor, the two reaction solutions collide and combine outside the reactor, thereby avoiding microchannel obstruction issues and facilitating a quicker and more convenient synthesis process. This study examined the effect of the jet flow rate and dispersion addition on the morphology and size of silver powder particles. Based on the results of this study, spherical and dendritic silver particles with a rough surface can be prepared by adjusting the flow rate of the reaction solution and gelatin concentration. The microjet flow rate of 75 mL/min and the injected gelatin amount of 1% of the silver nitrate mass produced spherical ultrafine silver particles with a size of 4.84 μm and a tap density of 5.22 g/cm^3^.

## 1. Introduction

Silver is widely used in various industrial fields due to its good electrical conductivity, thermal conductivity, and ductility. As a functional material, silver at the micronano level shows a structure between crystalline and amorphous states, exhibits a modified surface molecular arrangement and crystal structure, and features enhanced surface activity [1]. The ultrafine silver powder shows spherical (or quasispherical), flake-like, dendritic, and microcrystalline morphologies [2,3,4]. Additionally, the micronano silver powder exhibits excellent performance in the fields of sound, light, electricity, magnetism, heat, and catalysis due to its small particle size, large specific surface area, high surface activity, and good catalytic activity. Furthermore, silver powder has antibacterial and sterilization capabilities because of its adsorption capacities and excellent optical properties due to its surface plasmon resonance. These characteristics have expanded the application of silver powder to fields such as electronics, the chemical industry, medicine, aerospace, the military industry, and metallurgy [5,6].

Various methods can be used to prepare silver powders, including physical, chemical, and biological methods [7]. Common physical preparation methods include the high-energy ball-milling method [8], spray thermal decomposition method [9], plasma evaporation condensation method [10], liquid phase reduction method [11,12], microemulsion method [13,14], liquid–solid phase reduction method [15], and microbial reduction [16]. The chemical method is widely used in large-scale industrial production because of its low equipment requirements and energy consumption.

Microchemical reactors mainly provide controllable and high-throughput chemical synthesis methods with good stability, low energy consumption, small reaction volume, and uniform reaction conditions [17,18,19]. Therefore, they provide a new process strategy for materials science, chemical synthesis, biomedical diagnosis, and drug screening [20,21,22]. Fouling (i.e., unnecessary deposition on the surface) often occurs and causes local constriction in microstructure equipment, which changes the flow rate and increases the pressure drop or even completely blocks the microchannel. Hence, this is the biggest obstacle to the effective operation of microstructure equipment [23]. Rathi [24] optimized the continuous synthesis of crosslinked chitosan sodium tripolyphosphate (CS-TPP) nanoparticles using a microreactor and compared it with a batch-stirred reactor. Lim [25] described a straightforward and adaptable coaxial turbulent jet mixer that not only synthesized various nanoparticles (NPs) at high throughput but also maintained the benefits of homogeneity, reproducibility, and tunability that could typically be attained only in specialized microscale mixing equipment. Using various conditions, Sebastian [26] obtained complex metal nanomaterials, such as Pt–Pd heterostructures, Ag–Pdcore–shell NPs, and Au–Pd dumbbell structures and achieved fine control of material size and morphology using the homogeneous microfluidic reactor. Baber [27] investigated AgNO_3_ reduction by NaBH4 in an impinging jet reactor (IJR) to prepare silver NPs. Under certain conditions, the size of the silver NPs could be controlled at 4.3 ± 1 nm and 4.7 ± 1.3 nm. Sahoo [28] reported that the small size and uniformity (5.2 ± 0.9 nm) of silver NPs can be controlled using a free impinging stream reactor at room temperature. The unique IJR characteristics are effective mixing and the lack of channel walls to avoid fouling. 

The Y-type microjet reactor makes the reaction solution converge outside the reactor; therefore, the solutions are uniformly mixed and reacted. The process is safe, efficient, and controllable, as required by modern chemical technologies, while effectively avoiding precipitation blockage problems in the microchannel. In this study, the Y-type microjet reactor was used to make two reactant solutions that are uniformly mixed and reacted at room temperature (23 °C ± 2 °C). Additionally, the microjet method’s effects on micron silver powder’s morphology, particle size, and dispersion performance were investigated by controlling the gelatin amount in the system and the jet flow rate. It is hoped that silver powder’s morphology and particle size can be controlled within a certain range.

## 2. Materials and Methods

### 2.1. Materials

Gelatin (industrial gelatin) was purchased from Shanghai Maclean Biochemical Technology Co., Ltd. (Shanghai, China) Silver nitrate (AgNO_3_) was purchased from Tongbai Hongxin New Material Co., Ltd. (Henan, China) Ascorbic acid (C_6_H_8_O_6_) was purchased from Zhengzhou Tuoyang Industrial Co., Ltd. (Zhengzhou, China) Sodium chloride (NaCl), nitric acid (HNO_3_), and absolute ethanol (C_2_H_6_O) were purchased from Chengdu Kelong Chemical Co., Ltd. (Chengdu, China) Water used in this study was deionized. All the reagents were of analytical purity and were used without additional purification.

### 2.2. Experimental Methods

The experimental device for preparing silver powder particles, the Y-type microjet reactor, is shown in Figure 1. The Y-type microjet reactor is made of 3D-printed photosensitive resin. The channel’s inner diameter is 1.0 mm, the distance between the two outlets is d = 10 mm, and the jet’s intersection angle is 45°.

Ascorbic acid and silver nitrate undergo the following chemical reduction reaction:2AgNO_3_ +C_6_H_8_O_6_ = 2Ag↓+C_6_H_6_O_6_ +2HNO_3_(1)

The synthesis procedure and other experimental conditions for the silver particles’ preparation in this study are shown in Figure 2. Solutions (A and B) were prepared as follows: A certain amount of AgNO_3_ and C_6_H_8_O_6_ were dissolved in deionized water, and an appropriate amount of HNO_3_ was added to adjust the pH value of the C_6_H_8_O_6_ solution. Further, gelatin (1.0–3.0% of the AgNO_3_ mass) was added to the C_6_H_8_O_6_ solution as a dispersant.

The prepared solutions A and B were delivered to the two inlets of the microjet mixing reactor by advection pumps, thereby providing appropriate flow rates and producing the desired jets at the two outlets. When the two jets collide, the solvents mix and subsequently react. The mixed solutions were poured vertically into a lower beaker filled with 100 mL of deionized water and rotated at 200 rpm. The spraying of the two solutions was arranged to ensure that the total volume of the final solution was 200 mL and the solution in the beaker was stirred for 30 s. Following the reaction, the solution stratified and precipitated after standing. The supernatant was removed and mixed with a 10% NaCl solution without white flocculent precipitation, which revealed that the silver nitrate had been totally reduced. The reaction was conducted at room temperature (23 °C ± 2 °C); all concentrations stated are those of the inflow before reagent mixing. The layered solution was filtered, washed, and dried before yielding the silver powders.

### 2.3. Characterization Testing

The morphology of silver powders was investigated using Nova Nano SEM450 field emission scanning electron microscopy (SEM, American FEI Company, Hillsboro, OR, USA). The physical phases of the silver powders were characterized by X-ray diffraction (XRD, Xpert powder, PANalytical, Amsterdam, The Netherlands). The particle size distribution of the silver powder was determined using a laser particle size meter (Rise-2002, Jinan Runzhi Technology Co., Ltd., Jinan, China). The specific surface area of silver powders was tested with BET-specific surface area measurement (DX 400, Beijing Jingwei Gaobo Science and Technology Co., Ltd., Beijing, China).

## 3. Results and Discussion

### 3.1. Effect of Preparation Method on the Morphology of Silver Powders

The silver powder prepared by the conventional method, and the microjet reactor was characterized by XRD, and the results are shown in Figure 3.

As seen in Figure 3, the spectral lines have characteristic peaks at 38.1°, 44.23°, 64.37°, 77.36°, and 81.46°, which correspond to the (111), (200), (220), (311), and (222) crystal planes of cubic crystalline silver, respectively. Furthermore, these spectral lines are consistent with the monolithic silver standard pattern (JCPDS 04-0783). There were no other diffraction peaks in the spectrum, and the diffraction peaks of the curve were quite sharp. This indicates that the silver powder products obtained by the two experimental methods were highly crystalline and comprised monolithic silver.

The gelatin addition of 1% was chosen in the conventional approach to configure the AgNO_3_ solution and the C_6_H_8_O_6_ solution under the same other conditions. The AgNO_3_ solution was added into a beaker containing C_6_H_8_O_6_ mixed solution. The reaction solution was rinsed three times with deionized water and anhydrous ethanol before being dried at 60 °C for 4–6 h to obtain the silver powder product.

The SEM images of the silver powder prepared using the conventional and microjet methods are shown in Figure 4.

Figure 4a demonstrates that the silver powder particles prepared using the conventional method were not agglomerated and were monodispersed and polyhedral in shape. The silver powder particle size was about 2–5 μm, and the surface was smoother than the silver particles prepared using the Y-type microjet reactor. Figure 4b shows that the silver particles prepared using the Y-type microjet reactor aggregated from smaller particles into large spherical particles. Moreover, the silver powder morphology was mostly spherical with a rough surface.

### 3.2. Effect of Dispersant Dosage on the Morphology under Microjet Conditions

The microjet flow rate effect on the silver powder morphology was investigated at different contents of gelatin addition.

The flow rates of the microjet reactor were set at 50, 75, and 100 mL/min. A sufficient amount of gelatin was adsorbed on the silver particles’ surface, successfully preventing the particles from adhering together. The silver powder was produced by mixing the reactant solutions (A and B) at a certain gelatin dispersion amount (1%, 1.5%, and 2%). Other conditions were consistent with the description of the experimental procedure presented in Section 2.2. Prior research has demonstrated that the amount of mixing between the two reactant solutions as well as the morphology and size of the micron and nanoparticles formed as a result of chemical reduction were all considerably influenced by the flow rate of the microjet reactor [27].

The high-magnification SEM images of the silver powder prepared under different jet flow rates by adding 1% gelatin are shown in Figure 5. Furthermore, Figure 5a demonstrates that when the solution flow rate was 50 mL/min, the formed silver powder particles contained both a sizable number of symmetrical dendritic particles as well as near-spherical particles with rough surfaces. When the flow rate was 75 mL/min, the formed silver powder particles were spherical particles with a rough surface. When the solution flow rate was 100 mL/min, the prepared silver powder particles were spherical particles with a rough surface, similar to that observed in the case of the particles in Figure 5b.

The high-magnification SEM images of the silver powder prepared using the Y-type microjet reactor by adding 1.5% gelatin while keeping other experimental conditions constant is shown in Figure 6.

Figure 6 shows that the morphology of the silver powder changed considerably with an increase in the solution flow rate. It can be seen from Figure 6a that when the solution flow rate was 50 mL/min, the prepared silver powder particles were mostly spherical with a few dendritic particles. However, when the solution flow rate was 75 mL/min, the formed silver powder particles were mostly dendritic with a certain thickness and fewer sphere-like particles (Figure 6b). It can be observed from Figure 6c that when the solution flow rate was 100 mL/min, the formed silver powder particles were almost dendritic with very few spherical particles. However, compared with Figure 5a (the 1.5% gelatin addition), the branches of dendritic particles formed in Figure 6b were wider and thicker.

Figure 7 shows the high-magnification SEM images of the silver powder fabricated with the microjet method under the same experimental circumstances except that here, 2% gelatin is added.

Under this condition, the change in silver powder morphology was not obvious with an increase in the solution flow rate. When the solution flow rate was varied between 50, 75, and 100 mL/min, the morphological changes in the synthesized silver powder particles were not obvious; moreover, all of them were irregular dendrites with irregular particle surfaces. The dendritic particles formed by adding 2% gelatin had wider branches and rougher surfaces than those prepared with less gelatin addition (Figure 5 and Figure 6).

### 3.3. Effect of Dispersant Dosage on the Particle Size of Silver Powder under Microjet Conditions

The effect of the microjet flow on the particle size of silver powder was investigated by adding different amounts of gelatin. The microjet reactor’s flow rates were set at 50, 75, and 100 mL/min. Similarly, the silver powder was prepared by mixing the reactant solutions (A and B) with three different gelatin amounts (1, 1.5, and 2%) using the Y-type microjet reactor, and other conditions were consistent with the description of the experimental procedure in Section 2.2.

Figure 8d displays the low magnification SEM pictures, distribution map, and diameter D_50_ distribution map of the silver powder created at different jet flow rates with a 1% gelatin addition. When the gelatin addition was 1%, it can be seen that the particle size distribution of the silver powder did not change much with the modification of the jet flow rate. As shown in Figure 8d, the jet flow rates of 50, 75, and 100 mL/min correspond to the diameters D_50_ of the silver powder of 4.88, 4.84, and 5.35 μm. This demonstrates that when the amount of gelatin added is 1%, jet flow has little effect on the particle size of silver powder.

The low-magnification SEM images, particle size distributions, and D_50_ diameter distributions of the silver powder prepared under different jet flow rates by adding.5% gelatin are shown in Figure 9. From Figure 9d, it can be seen that the particle size of silver powder particles gradually increases with an increase in the microjet flow rate. Figure 9a shows that when the jet flow rate was low (50 mL/min), the silver powder particles were dispersed and had smaller particle sizes compared to Figure 8b,c.

Therefore, when the gelatin addition was 1.5%, the sphericity of the synthesized silver powder particles decreased. The particle size increased considerably when the microjet flow rate was higher. When the jet flow rate was too high, irregular and flaky silver powder particles with large particle sizes and low dispersion were formed.

The low-magnification SEM images, particle size distributions, and D_50_ diameter distributions of the silver powder prepared under different jet flow rates by adding 2% gelatin are shown in Figure 10. It is obvious from the figure that when too much gelatin was added (2%), the silver powder particles had poor dispersion and large particle sizes, and the maximum D_50_ diameter size of particles reached 17.75 μm.

However, excessive gelatin will make the reaction liquid viscous, thereby decreasing the contact area of ascorbic acid and silver nitrate and slowing down the reaction rate. Simultaneously, the elemental silver formation also slows down because of the thicker gelatin film diffusion. This results in particle nucleation and growth from the two phases that cannot be effectively separated, and the resulting silver powder would have larger particle sizes [29]. Moreover, the amount of gelatin used is too large and inconvenient for washing and filtration later. The microreactor has a high jet flow rate when using a Y-type microjet reactor to prepare silver powder particles. When impinging and mixing, faster formation of silver crystal nuclei occurs but not fast enough to consume all silver ions. Hence, there will still be regular crystal growth, forming a large number of dendritic silver crystals. Therefore, the added dispersant gelatin amount should not exceed 1% of the AgNO_3_ mass, and the jet flow rate should not be too high when preparing silver powder particles by impinging the jet method.

### 3.4. Effect of Dispersant Addition on Silver Powder Parameters

A potential synthesis mechanism of silver particles by conventional and microjet methods is shown in Figure 11. The morphology of the silver powder produced by various preparation techniques was quite diverse, as can be seen from the image. Ag^+^ in the solution was gradually converted to silver atoms by adding the reducing agent ascorbic acid. The silver atoms developed into polyhedral, spherical, and dendritic particles due to the addition of gelatin and the mixing method of reaction solutions.

Figure 12 demonstrates the connection between the added gelatin amount and the D_90_/D_10_ ratio, tap density, and specific surface area of the manufactured silver powder particles.

A larger D_90_/D_10_ ratio indicates a wide particle size distribution and low dispersion [30]. The D_90_/D_10_ ratio of the six samples was low, and the greatest value achieved was 4.24, as shown in Figure 12a. The D_90_/D_10_ ratio essentially exhibited a decreasing trend when the amount of gelatin was added. When the gelatin amount injected was fixed, a slightly higher jet flow rate could result in a smaller D_90_/D_10_ ratio.

As can be observed from Figure 12b, the maximum tap density (5.46 g/cm^3^) was obtained when 1% gelatin was added. When the tap density of silver powders was high, the crystallinity was enhanced, the buildup between the silver powder particles in the natural state showed enhanced density, and the void ratio was small. When it is used for silver paste and other applications, the conductive film obtained after sintering the slurry has fewer and smaller voids, the resistance of series in the circuit is small, and the electrode conductivity is excellent [31]. The tap density of the silver powder particles fell as the gelatin amount increased. When 1% gelatin was introduced, the tap density increased as the jet flow rate increased. When 2% and 3% of gelatin were applied, the tap density decreased as the jet flow rate rose.

The morphology of the silver particles changed from spherical to dendritic as more gelatin was added, and the specific surface area gradually expanded, as illustrated in Figure 8c. As the particular surface area increased, the surface activity of silver powder increased. The maximum specific surface area of the produced silver particles was 1.43 m^2^/g when the gelatin content was 2%. When 1% gelatin was added, the specific surface area of the silver powder particles gradually reduced with an increasing flow rate, although the difference was not considerable. Alternatively, the specific surface area of the silver powder particles gradually increased at 2% and 3% of added gelatin.

## 4. Conclusions

(1)The outcomes of this study demonstrate that under specific circumstances and within a specific range, the Y-type microjet reactor may be utilized to regulate silver particles’ morphology and particle size.(2)By changing the experimental conditions, spherical and dendritic silver particles can be obtained using a Y-type microjet reactor. When the microjet flow rate was 75 mL/min, and the gelatin content was 1% of the AgNO_3_ mass, the ultrafine spherical silver powder with a particle size of 4.84 μm and a tap density of 5.22 g/cm^3^ could be synthesized using the microreactor at room temperature.(3)Compared to conventional stirred reactors, the Y-type microjet reactor can quickly and efficiently mix reactant solutions, and the process is controllable. This reactor, unlike other microchemical reactors, does not have synthetic product deposition or channel blockage problems. The controlled synthesis of silver nanoparticles offers a potential future application for the microjet reactor.

## Figures and Tables

**Figure 1 materials-16-02217-f001:**
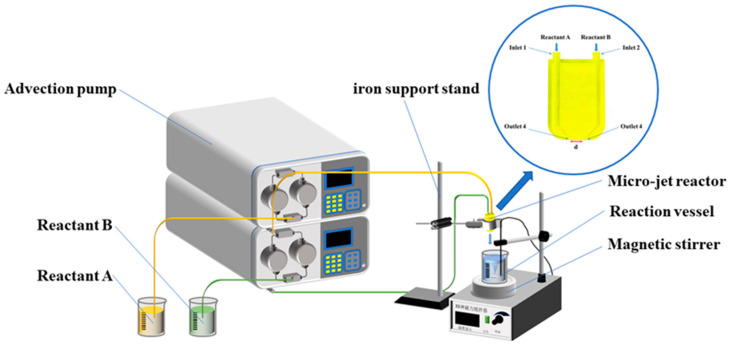
Schematic flow diagram of the reaction device.

**Figure 2 materials-16-02217-f002:**
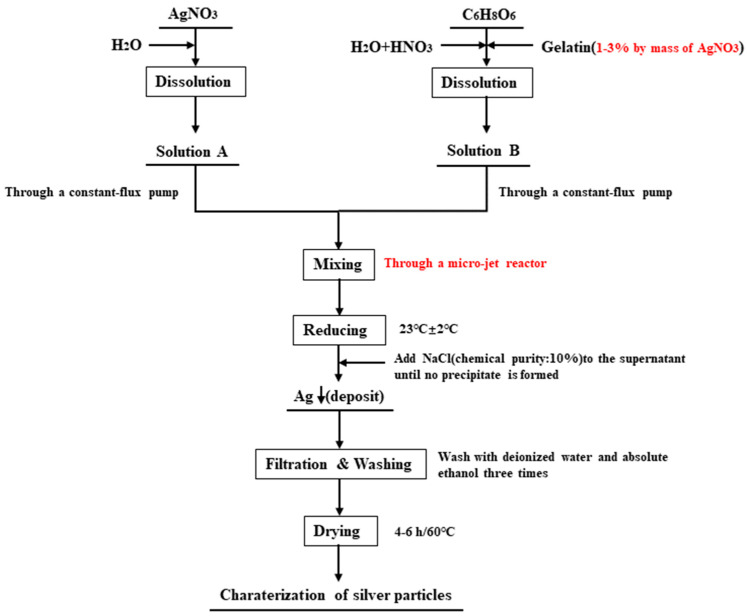
Experimental flow chart of silver particle preparation.

**Figure 3 materials-16-02217-f003:**
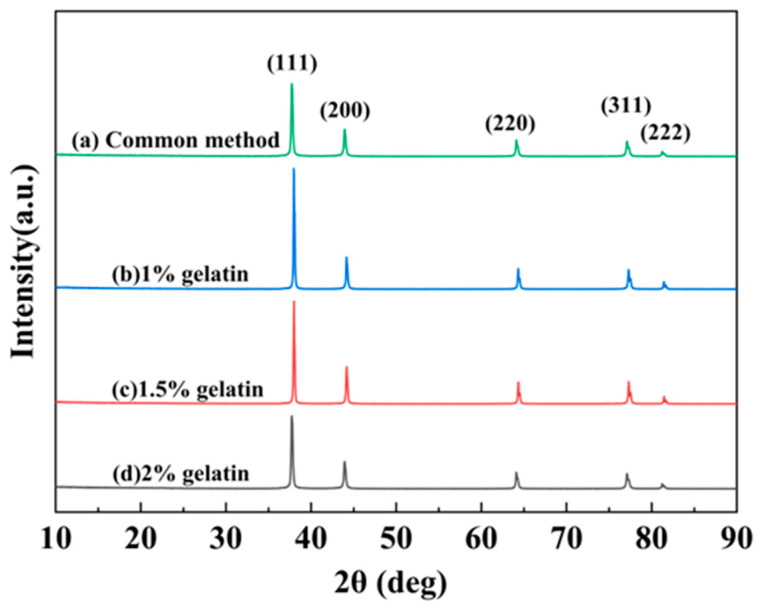
XRD patterns of silver powders were obtained using different preparation methods. (a) Conventional method; (b) microinjection method by adding 1% gelatin; (c) microinjection method by adding 1.5% gelatin; and (d) microinjection method by adding 2% gelatin.

**Figure 4 materials-16-02217-f004:**
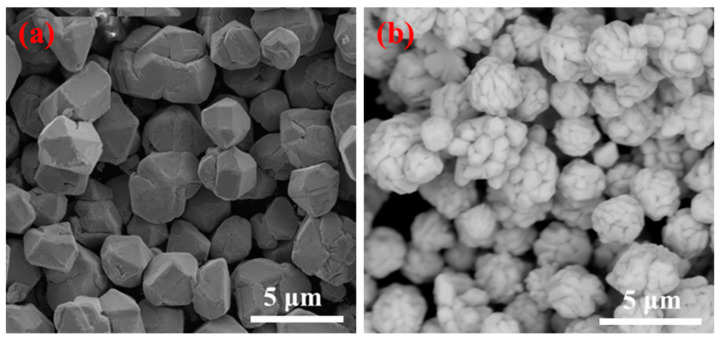
SEM of silver particles prepared by the conventional drop-in method. (**a**) Conventional method and (**b**) microjet method.

**Figure 5 materials-16-02217-f005:**
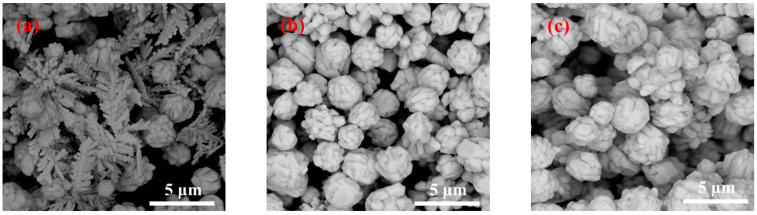
SEM images of the silver powder prepared under different flow rates by adding 1% gelatin. (**a**) 50 mL/min; (**b**) 75 mL/min; and (**c**) 100 mL/min.

**Figure 6 materials-16-02217-f006:**
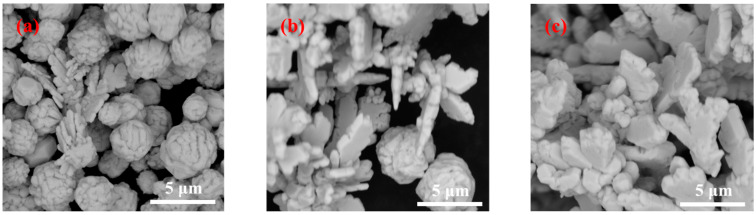
SEM images and particle size distribution of silver powders prepared under different flow rates by adding 1.5% gelatin. (**a**) 50 mL/min; (**b**) 75 mL/min; and (**c**) 100 mL/min.

**Figure 7 materials-16-02217-f007:**
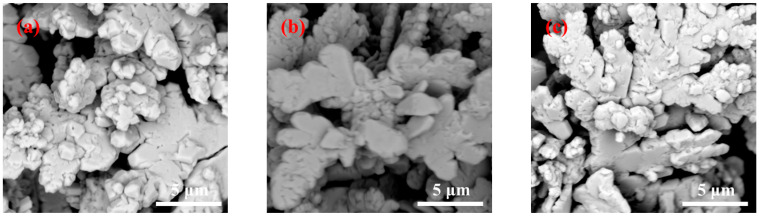
SEM images and particle size distribution of silver powders prepared under different flow rates by adding 2% gelatin. (**a**) 50 mL/min; (**b**) 75 mL/min; and (**c**) 100 mL/min.

**Figure 8 materials-16-02217-f008:**
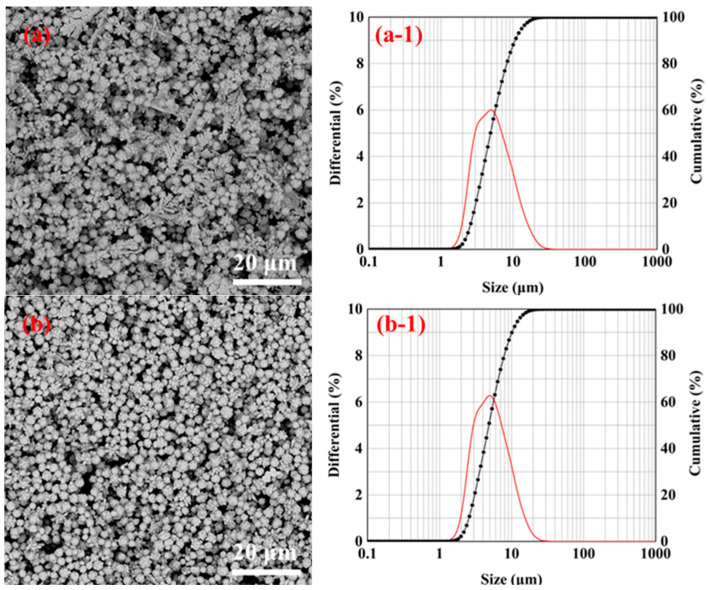
SEM images, particle size distributions, and D_50_ distributions of the silver powder prepared by adding 1% gelatin under different jet flow rates. (**a**) 50 mL/min; (**b**) 75 mL/min; (**c**) 100 mL/min; and (**d**) D_50_ distribution of silver powder.

**Figure 9 materials-16-02217-f009:**
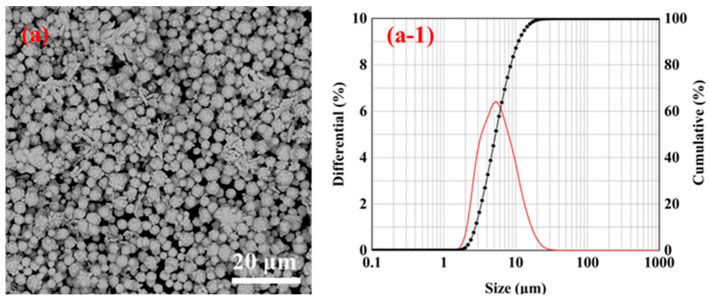
SEM images, particle size distributions, and the D_50_ distributions of silver powders were prepared by adding 1.5% gelatin and under different jet flow rates. (**a**) 50 mL/min; (**b**) 75 mL/min; (**c**) 100 mL/min; and (**d**) D_50_ distribution of silver powders.

**Figure 10 materials-16-02217-f010:**
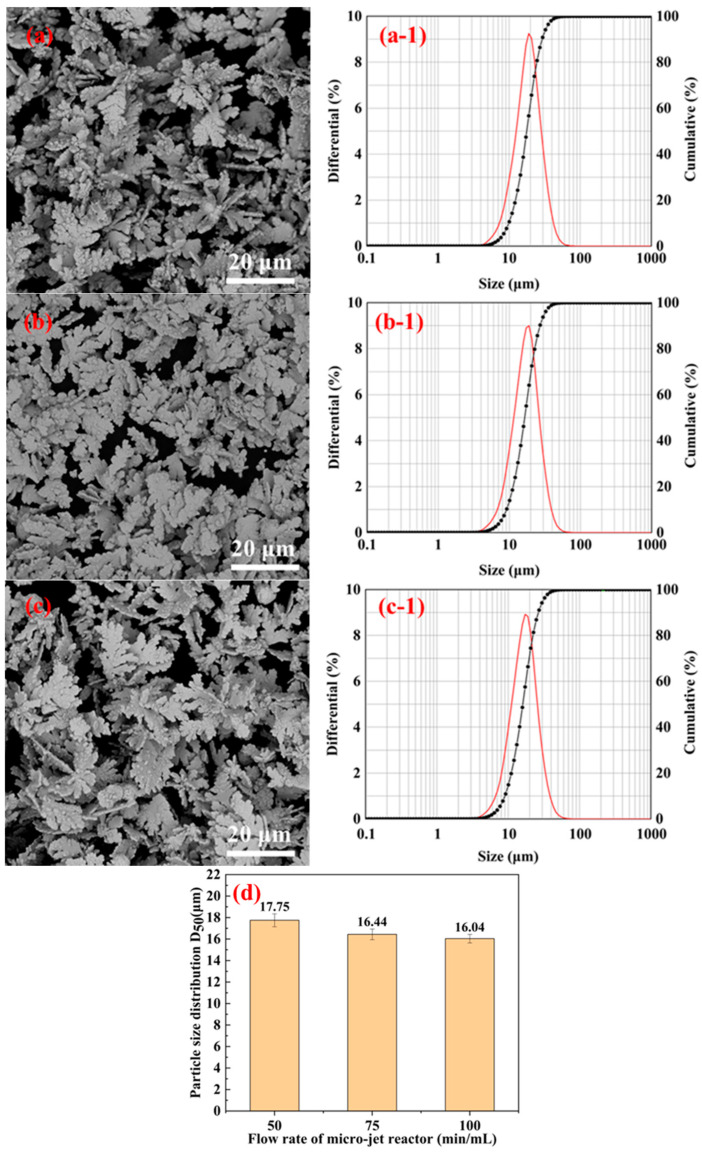
SEM images, particle size distribution, and D_50_ distribution of silver powders prepared under different jet flow rates by adding 2% gelatin. (**a**) 50 mL/min; (**b**) 75 mL/min; (**c**) 100 mL/min; and (**d**) D_50_ distribution of silver powder.

**Figure 11 materials-16-02217-f011:**
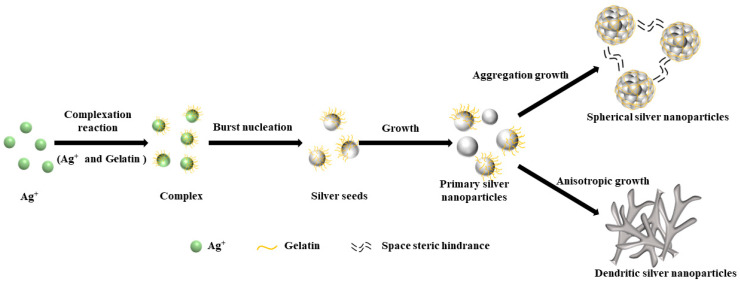
A schematic diagram of the growth process of spherical and dendritic micron silver particles.

**Figure 12 materials-16-02217-f012:**
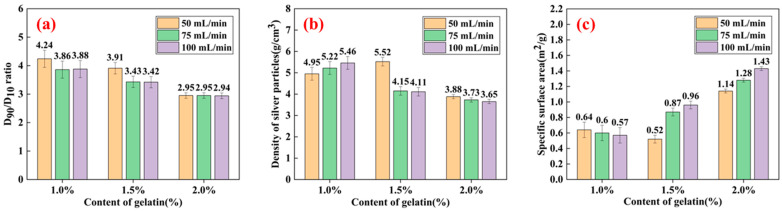
Properties of the silver powders prepared using different gelatin additions and jet flow rates. (**a**) D_10_/D_90_; (**b**) density of silver particles; and (**c**) specific surface area.

## Data Availability

The data presented in this study are available in this article.

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
