# Peer review of "Preparation of Spherical Ultrafine Silver Particles Using Y-Type Microjet Reactor"

_materials, 2023, doi:10.3390/ma16062217_

Round 1
Reviewer 1 Report
Please see the comments.

Reviewer 2 Report
Reviewer general comment:
The authors have Designed and executed the experiment very well. In fact, the manuscript titled Preparation of spherical ultrafine silver particles by Y-type micro-jet reactor is also written in a good way. However, the authors failed to explain and draw out the novelty of the work, this aspect needs to be improved. This work is worthwhile to be publish in this journal after major revision. The following issues should be addressed:
Reviewer comment 1:
English language must be revised in the manuscript.
Reviewer comment 2:
Introduction is well-organized but the importance and novelty of the research should be highlighted and more clearly stated. The authors should give some examples of works in the bibliography, to clear the advantage of their work in comparison with those works.
Reviewer comment 3:
Abstract and conclusion not targeted; the authors should rephrase it.
Reviewer comment 4:
In materials section Gelatin type should be clarified
Reviewer comment 5:
Authors should provide a mechanism of synthesis of silver nanoparticles.
Reviewer comment 6:
In the discussion part Compare the result of using a ascorbic acid as reductive material with other reducing agent used to prepare spherical Ag nanoparticles.
Reviewer comment 7:
in line 163 are you sure they are spherical?

Reviewer 3 Report
Present article is devoted to the synthesis of Ag nanoparticles, which is already studied by many authors. The originality of the present work deals with usage of jet reactor. It is highly like, that the results obtained will be interesting for the scientists working on chemical technology and engineering.
English language should be polished.
The significance of the results can be considered to be sufficient for the publication, all conclusions are supported by experimental data.
Paper requires minor revision.
The following points should be corrected or described in detals:
1. There is now information about the density measurements for the powders in Experimantal section
2. For BET measurements the surface areas of 0.57 and 0.60 seems almost identical. What the statistics say?
Round 2
Reviewer 1 Report
In this revised version of the manuscript, the authors have responded in a separate document to my criticism. So, it could be accepted.